# A Bioinformatics Evaluation of the Role of Dual-Specificity Tyrosine-Regulated Kinases in Colorectal Cancer

**DOI:** 10.3390/cancers14082034

**Published:** 2022-04-18

**Authors:** Amina Jamal Laham, Raafat El-Awady, Jean-Jacques Lebrun, Maha Saber Ayad

**Affiliations:** 1Sharjah Institute for Medical Research, University of Sharjah, Sharjah 27272, United Arab Emirates; amina.laham@mail.mcgill.ca; 2College of Medicine, University of Sharjah, Sharjah 27272, United Arab Emirates; 3College of Pharmacy, University of Sharjah, Sharjah 27272, United Arab Emirates; 4Cancer Research Program, Department of Medicine, McGill University Health Center, Montreal, QC H4A 3J1, Canada; jj.lebrun@mcgill.ca

**Keywords:** colorectal cancer, dual-specificity tyrosine-regulated kinases (DYRKs), DYRK1A, kinases, targeted therapy, bioinformatics

## Abstract

**Simple Summary:**

The dual-specificity tyrosine-regulated kinase (DYRK) family has been implicated in various diseases, including cancer. However, its role in colorectal cancer has not been elucidated. In this research, we used publicly available web-based tools to investigate DYRKs status in colorectal cancer. Our results showed that among *DYRKs*, only *DYRK1A* was upregulated significantly in late tumor stages, and it is associated with poor prognosis for colorectal cancer patients. These finding comprehensively characterized DYRK1A as a potential new therapeutic approach in CRC, especially in late tumor stages.

**Abstract:**

Colorectal cancer (CRC) is the third most common cancer worldwide and has an increasing incidence in younger populations. The dual-specificity tyrosine-regulated kinase (DYRK) family has been implicated in various diseases, including cancer. However, the role and contribution of the distinct family members in regulating CRC tumorigenesis has not been addressed yet. Herein, we used publicly available CRC patient datasets (TCGA RNA sequence) and several bioinformatics webtools to perform in silico analysis (GTEx, GENT2, GEPIA2, cBioPortal, GSCALite, TIMER2, and UALCAN). We aimed to investigate the *DYRK* family member expression pattern, prognostic value, and oncological roles in CRC. This study shed light on the role of distinct DYRK family members in CRC and their potential outcome predictive value. Based on mRNA level, *DYRK1A* is upregulated in late tumor stages, with lymph node and distant metastasis. All DYRKs were found to be implicated in cancer-associated pathways, indicating their key role in CRC pathogenesis. No significant *DYRK* mutations were identified, suggesting that *DYRK* expression variation in normal vs. tumor samples is likely linked to epigenetic regulation. The expression of *DYRK1A* and *DYRK3* expression correlated with immune-infiltrating cells in the tumor microenvironment and was upregulated in MSI subtypes, pointing to their potential role as biomarkers for immunotherapy. This comprehensive bioinformatics analysis will set directions for future biological studies to further exploit the molecular basis of these findings and explore the potential of DYRK1A modulation as a novel targeted therapy for CRC.

## 1. Introduction

Colorectal cancer (CRC) is the third most common diagnosed type of cancer and represents the third leading cause of cancer-related deaths worldwide [1]. The vast majority of colorectal cancer patients are diagnosed at late stages and often exhibit poor prognosis with an estimated 5-year survival rate of less than 10% for stage 4 metastatic CRC patients [2]. The standard of care for stage 4 metastatic CRC includes surgery and chemotherapy; the latter improves the 5-year survival rate by 30–50% [3,4]. However, patients will eventually develop resistance to chemotherapy and often develop tumor relapse following surgery [5]. Extensive research over the past 20 years defined important prognostic biomarkers and therapeutic targets in CRC, further improving the survival of CRC patients.

One of the most mutated genes in cancer are protein kinases. The dysregulation of kinases is found to be an early event in carcinogenesis [6]. Kinases are classified into different group, and one of the major kinases groups is the proline-directed kinases “(CMGC)” group, which includes cyclin-dependent kinase (CDK), mitogen-activated protein kinase (MAPK), glycogen synthase kinase (GSK3), CDC-like kinase (CLK)) and dual-specificity tyrosine-regulated kinase (DYRK) [7].

As potential targets for colorectal cancer treatment, protein kinases have attracted remarkable attention in recent years and many such kinases have been found to be implicated in CRC initiation, progression, and/or metastasis [6]. During the last decade, a lot of attention has been focused on MAPK/ERK and PI3 Kinase pathways due to their broad involvement in tumorigenic processes in various tissues [8]. These studies led to the development of several targeted therapies, such as cetuximab and panitumumab, which successfully improved the survival of metastatic CRC patients [9]. Besides MAPK and PI3K, the dual-specificity tyrosine-regulated kinase (DYRK) family has also been found to be implicated in regulating tumorigenesis [10]. In humans, there are five DYRK family members, which are classified into class 1 (DYRK1A and DYRK1B) and class 2 (DYRK2, 3, and 4) based on sequence homology of their kinase domain [11]. DYRKs have the ability to auto-phosphorylate their own activation loop on tyrosine residue while still attached to the ribosome during translation. This phosphorylation event has been reported to be essential for DYRK kinase activity such as serine and threonine kinases [12]. DYRK family members play essential cellular functions, and the dysregulation of their kinase activity was found to be implicated in various diseases, such as neurological disorders, metabolic disorders and cancer [11,13,14]. Class 1 DYRKs (1A and 1B) were found to be essential for cell cycle regulation in neurological, mesenchymal and epithelial tissues [15,16,17,18]. Studies have reported class 1 DYRKs to be involved in tumorigenesis, but their precise role and functions remain unclear (10,19). DYRK1A can exert tumor-suppressive activities in cancer and was found to regulate cancer-related key pathways, such as apoptosis, DNA damage, activation of receptor tyrosine kinase and angiogenesis [19]. However, DYRK1A expression levels vary among different types of cancer. For example, it was found to be upregulated in some solid cancers, such as glioblastoma, lung, head and neck, and pancreatic cancers [20,21,22]. Recently, it was also reported to be upregulated in B cell acute lymphoblastic leukemia [23]. In contrast, it was found to be downregulated in acute myeloid leukemia [24]. Unlike DYRK1A, DYRK1B was reported to exert pro-oncogenic functions in ovarian, pancreatic, breast tumors and liposarcoma [25,26,27]. DYRK1B expression is also increased upon the inhibition of MAPK and PI3K/AKT/MTORC pathways in ovarian and pancreatic tumors, suggesting its involvement in the resistance to targeted therapies in these tumors [28,29]. From the class 2 DYRK subgroup, only DYRK2 was extensively studied in terms of tumorigenesis and found to have a controversial role in cancer [30]. DYRK2 is overexpressed in various cancer types [10] and is involved in the proteo-stasis pathway that can enhance cancer cells survival under stressful conditions [31]. However, other studies reported that it could exert a tumor suppressor role through inhibiting epithelial-to-mesenchymal transition (EMT) in breast and ovarian cancers [32,33]. Furthermore, DYRK2 was also reported to suppress liver metastasis and to act as a good prognostic marker in CRC [34,35]. A few studies have been conducted to investigate the role of DYRK3 and 4 in cancer. The cellular role of DYRK3 was in fact recently discovered [36,37]. It has been reported to activate mTORC1 in stress granules and to allow cytoplasmic compartmentalization via liquid phase transitions [36]. Additionally, it was reported to be essential for mitosis; acting as a regulator for dissolution or condensation of membrane organelles during mitosis [37]. In cancer, DYRK3 was proposed to act in parallel with DYRK1A to promote cell survival under stressful conditions by phosphorylating nicotinamide adenosine dinucleotide (NAD)-dependent deacetylase SIRT1 [38].

Despite the important roles played by DYRK family members in various types of cancers with the exception of DYRK2, no study investigated their role and broad contribution to colorectal cancer. Recently, bioinformatics-based analysis of publicly available databases has gained popularity in biomedical research due to their robustness in identifying key cancer vulnerabilities and potential biomarkers. The available pathological-sample database provides a large RNA seq and proteomic datasets that can be instrumental in investigating the specific role and function of key regulatory genes in multiple types of cancer [39]. Moreover, they can provide invaluable information when the analysis is conducted in view of the clinical data including disease staging.

In this study, we analyzed DYRKs expression levels in tumor versus normal samples and in the different CRC molecular subtypes, in correlation with clinical features of CRC and patient survival outcomes. For this, we used various bioinformatics tools that are publicly available, including Genotype-Tissue Expression (GTEx), Gene Expression database of Normal and Tumor tissues 2 (GENT2), Gene Expression Profiling Interactive Analysis2 (GEPIA2), cBioPortal, Gene Set Cancer Analysis (GSCALite), TIMER2, UALCAN, Kaplan–Meier and Cox regression analyses. The overall aim of this study is to address the role of DYRKs in CRC carcinogenesis and to evaluate their capacity as potential prognostic markers and/or therapeutic targets in CRC.

## 2. Materials and Methods

### 2.1. Genotype-Tissue Expression (GTEx)

To measure the expression level of DYRKs in normal colon tissues, we used the Genotype-Tissue Expression (GTEx) web tool (https://gtexportal.org, accessed on 6 April 2022), which provides RNA seq data to study the expression of specific genes in different non-diseased tissue. We chose the expression tab with multi-gene query. We queried *DYRKs* in tumor tissue of the sigmoid and transverse colon. The website provides TPM (transcript per million) for each gene.

### 2.2. Gene Expression Database of Normal and Tumor Tissues 2 (GENT2)

To measure the expression level of *DYRKs* in different CRC subtypes, we used the GENT2 database (http://gent2.appex.kr/gent2/, accessed on 5 April 2022) [40]. This tool uses NCBI GEO database that is derived from microarray platforms (Affymetrix U133A or U133Plus2) [40]. We queried each *DYRK* alone by searching its gene across the subtypes tab; we used this tool to investigate the expression of *DYRKs* in CRC molecular subtypes.

### 2.3. Gene Expression Profiling Interactive Analysis2 (GEPIA2)

To measure the *DYRKs* expression level in different CRC stages, we used the GEPIA2 web-based server. The data are derived from the TCGA RNA sequence dataset (http://gepia2.cancer-pku.cn/#index, accessed on 6 April 2022) [41]. We queried each *DYRK* alone by searching its gene across stages tab. Moreover, we used this tool to generate the overall survival Kaplan–Meier plots for each *DYRK* in CRC patient. Hazard ratios were calculated based on Cox regression analysis.

### 2.4. cBioPortal

To further analyze the *DYRKs* status in tumor samples, we used cBioPortal [42,43]. We selected the Colorectal Adenocarcinoma TCGA PanCancer dataset (592 samples). We chose the gene specific query and selected mRNA expression z-score relative to normal samples (Log RNAseq V2 RSEM). For each *DYRK*, we filtered the samples based on z-score. Samples with z-score < 1 were considered to have low expression, and samples with z-score >1 were considered to have high expression. For *DYRK2*, all z-scores were below −1 in this dataset, so we were unable to compare it with other group, so we excluded it from the comparison.

### 2.5. GeneSet Cancer Analysis (GSCALite)

Gene Set Cancer Analysis (GSCALite) is a web-based server that uses to investigate the status of genes in cancer patient’s datasets (http://bioinfo.life.hust.edu.cn/web/GSCALite/, accessed on 5 April 2022) [44]. Interestingly, it can predict the gene activity in a cancer-associated pathway; we used this tool to investigate the role of DYRKs in cancer-related pathways in a colon adenocarcinoma (COAD) data set from TCGA.

### 2.6. STRING

We used STRING web tool to predict functional protein interaction network in *Homo sapiens* for DYRKs (https://string-db.org, accessed on 6 April 2022). We customized the active interaction sources and selected only experimental, database and co-expression proteins.

### 2.7. TIMER2

TIMER 2 is a web tool that uses multiple algorithms to analyze and measure the tumor-infiltrating immune cells from TCGA dataset (http://timer.cistrome.org, accessed on 5 April 2022). This tool measures the correlation between the expression of the gene of interest and immune-infiltrating cells abundancy. We used this tool to investigate the correlation between DYRKs and tumor-infiltrating immune cells in CRC [45]. We selected immune association to each gene and each type of immune cells, and then we chose TIMER algorithm and Spearman correlation.

### 2.8. UALCAN

UALCAN is a web-based tool that uses OMICS data from cancer data sets such as TCGA, MET500 and CPTAC (http://ualcan.path.uab.edu, accessed on 6 April 2022). It is used to study the gene status in different cancers such as expression and methylation in correlation to survival and calculate significance by Student’s *t*-test [46]. We used this tool to investigate the expression of *DYRKs* in normal vs. tumor samples and to investigate *DYRKs* promoter methylation status in CRC TCGA data sets.

## 3. Results

### 3.1. DYRKs Expression in Colon Tissues

*DYRKs* are expressed ubiquitously in all human tissue, and they have low tissue specificity [11]. We used the GTEx portal web tool to investigate the expression level of *DYRKs* in normal colon tissues in two anatomical parts: the sigmoid and the transverse colon. As represented in Figure 1, *DYRK1A* was found to have a higher expression level in colon tissues compared to other *DYRKs* in both anatomical parts of the colon. *DYRK1B* and *DYRK2* showed intermediate expression levels, whereas *DYRK3* and *DYRK4* were found to have low expression in colon tissues.

### 3.2. DYRKs Genes Expression in Normal vs. Colon and Rectum Adenocarcinoma

Next, we measured the expression level of *DYRKs* in colon and rectum adenocarcinoma [COAD and rectal adenocarcinoma (READ)] vs. normal tissues. Using the RNA seq dataset from TCGA data in UALCAN web tool, we found no significant change in *DYRK1A* and *1B* (Figure 2a,b), whereas *DYRK2* (Figure 2c) had a significantly lower expression in COAD samples compared to normal tissues. On the other hand, both DYRK3 (Figure 2d) and *DYRK4* (Figure 2e) were found to be significantly upregulated in COAD samples vs. normal tissues. As shown in Figure 3, both *DYRK1A* (Figure 3a) and *DYRK2* (Figure 3c) showed a significantly lower expression in READ tumor vs. normal tissues, whereas *DYRK1B*, *DYRK3* and *DYRK4* had no significant change in expression.

### 3.3. DYRKs Expression Level in CRC Molecular Subtypes

CRC are classified into major pathological molecular subtypes. The first group is microsatellite instable group (MSI) due to defects in mismatch repair (MMR) genes (16%) or DNA polymerase epsilon proof reading (3%). The second group is microsatellite stable (MSS) and represents 84% of CRC cases. MSS CRC are characterized by high rate of somatic mutations in several genes such as *APC*, *TP53*, *PIK3CA*, *KRAS* and *SMAD4* [47]. To study *DYRK* members’ gene expression in the different tumor subtypes, we used the web-based tool GENT2 [40]. Comparing the MSI and MSS subtypes, *DYRK1A*, *DYRK3* and *DYRK4* (Figure 4a,d,e) are all significantly higher in MSI, whereas *DYRK2* is significantly higher in MSS (Figure 4c). For *DYRK1B*, there were no significant differences in its expression in MSI and MSS (Figure 4b).

### 3.4. DYRKs Expression Level in CRC Stages

Using GEPIA2, we investigated the level of *DYRKs* member in different CRC stages. As shown in Figure 5, *DYRK1A* is significantly upregulated in late tumor stages from IIIA to IVB (Figure 5a). *DYRK1B* expression did not show any significant differences among CRC stages (Figure 5b). Similarly, class 2 *DYRKs* expression did not have any significant differences in CRC stages. This indicates that among *DYRKs* members, only *DYRK1A* is overexpressed in later tumor stages of CRC patients and suggests that *DYRK1A* may play a role in the advanced stages of the disease.

### 3.5. Clinical Feature of DYRKs in CRC

To further explore the expression status of *DYRK* members in relation to the CRC clinical features, we used cBioPortal and filtered the patient data set based on z-score to low or high expression to each *DYRK* member.

#### 3.5.1. Histological Cancer Subtypes

Consistent with what was observed with our RNA seq from UALCAN web tool analysis (F2-F3), the expression level of *DYRK3* and 4 (z-score = or >1) were high in the majority of the tumor samples. Tumor samples with high *DYRK1A* were found to be significantly associated with mucinous colorectal adenocarcinoma cancer (Figure 6a). A similar pattern was obtained for *DYRK4*, as shown in Figure 6d. On the other hand, DYRK1B expression was higher in non-mucinous COAD than in mucinous subtype (Figure 6b). For *DYRK3*, there was no significant change in its expression in all three histological subtypes (Figure 6c).

#### 3.5.2. Metastasis, Lymph Node Stages and New Neoplasm Event Post Therapy

As shown in Figure 7, although *DYRK1A* was found to have a low expression in CRC compared to normal tissues, cancer samples with high *DYRK1A* expression were found to be enriched in late metastasis and lymph node stage (Figure 7a,b). For other *DYRKs* members, there were no statistical differences between their expression and metastasis and lymph node stages (Figure 7c–h). Moreover, tumor samples with higher *DYRK1A* level exhibited higher tumor recurrence following initial therapy (Figure 8a), whereas none of the other *DYRKs* members showed any significant change (Figure 8b–d). These results suggest that *DYRK1A* could serve as a potential negative tumor biomarker in the late stages of the CRC disease.

### 3.6. Mutations and Promotor Methylation of DYRKs in CRC

*DYRKs* have differential expression in normal vs. tumor samples in CRC. To identify if this variation in expression is related to specific mutations, we used the Pan TCGA data from cBioPortal and analyze the *DYRKs* mutations in this dataset. As shown in Figure 9, *DYRKs* in general are not mutated in CRC; some samples have amplification mutations in *DYRK1B*, *DYRK3* and *DYRK4*. For *DYRK1A*, there is a missense mutation with unknown significance. In general, the differential expression of *DYRKs* found to be related to mRNA level. Then, we used the UALCAN web tool to investigate the methylation of *DYRKs* genes promotor in different tumor stage. We found that *DYRK1A* methylation in COAD was not significantly changed (Figure 10a); however, in READ, the methylation level decreased significantly in stage 3 compared to normal samples (Figure 10b). The methylation of *DYRK1B* was found to be significantly increased in stage 1, 2, and 3 compared to normal in COAD (Figure 10c); in READ, the methylation increased significantly in stage 1 and decreased significantly in stage 4 compared to normal samples (Figure 10d). *DYRK2* was reported to have lower methylation level in stage 4 in COAD only (Figure 10e), and no significant change was observed in READ (Figure 10f). Investigating the mRNA expression in normal vs. tumor samples, *DYRK3* (Figure 10g) and *DYRK4* (Figure 10i) were found to have significantly lower methylation level compared to the control in COAD. In READ, only *DYRK4* was found to have lower methylation level in stage 1 and 4 (Figure 10i). These results suggest that variation in *DYRKs* expression in normal vs. tumors might be related to epigenetic regulation rather than gene mutations.

### 3.7. DYRKs Activity in Cancer Associated Pathways in CRC

After exploring the status of DYRKs in clinical features of CRC, we used GSCALite to predict the molecular function of DYRKs in COAD Figure 11a and in all cancers combined (Figure 11b). Class 1 DYRKs (1A and 1B) and DYRK4 were found to correlate with cell cycle inhibition. However, DYRK1B also correlated with the inhibition of apoptosis, DNA damage response, and PI3K/AKT pathway activation. DYRK2 was found to correlate with the inhibition of DNA damage response, whereas DYRK3 correlated positively with epithelial mesenchymal transition and estrogen hormone pathway. These data are predicted data based on gene correlation with major regulators of cancer-associated pathway and indicate that DYRKs may play a significant role in several cancer-related pathways. To further predict DYRKs functions in cancer, we used the protein functional association network tool (STRING) to predict the interacting partners of DYRKs. As shown in Appendix A, DYRK1A interacts with the DREAM complex that is involved in cell cycle inhibition. Moreover, it interacts with the ubiquitin ligase RNF169 that is involved in DNA damage response. These interactions were further demonstrated experimentally [48,49]. Moreover, DYRK1B was found to associate with Signal-induced proliferation-associated 1-like protein 1,2 and 3 (Appendix A), which are involved in morphogenesis and cytoskeletal organization. In correlation with the predicted cancer pathways, DYRK1B was also found to have a functional association with RNF169, which predicts its role in DNA damage. Another DYRK1B interactor is LZTS2, a negative regulator of beta catenin, suggesting its involvement in colorectal cancer [50]. In correlation with predicted cancer pathways, DYRK2 associated proteins were mainly DNA damage response factors such as P53 and ATM Appendix A. Finally, all DYRK members were found to have functional association with DCAF7, which was previously reported as an adaptor protein for DYRK1A [51].

### 3.8. Correlation between DYRKs and Tumor Infiltrating Immune Cells in CRC Patient

Because we found differential *DYRK* members expression in MSI and MSS subtypes and the fact that these two CRC subtypes exhibit different immune cells abundance in the tumor microenvironment [47], we used TIMER 2 web tool to explore potential correlation between *DYRKs* expression and tumor infiltrating immune cells in CRC.

As shown in Figure 12, *DYRK1A* significantly positively correlated with CD4^+^ T cells, macrophages and myeloid dendritic cells, with no significant correlation with CD8^+^ T cells and B cells. For *DYRK1B*, we only found a significant positive correlation with CD4^+^ T cells and macrophages. Although *DYRK2* has higher expression in the MSS subtype, its expression positively correlated with CD8^+^ T cells, CD4^+^ T cells, macrophages, and myeloid dendritic cells. This can be explained by the fact that the TIMIR web tool calculates the correlation in all CRC patient data sets not in a specific subtype. In addition, *DYRK3* was also found to be positively correlate with CD8^+^ T cells, CD4^+^ T cells, macrophages, and myeloid dendritic cells. *DYRK4* did not show significant correlation with any of the immune infiltrating cells. In conclusion, only *DYRK1A* and *DYRK3* are upregulated in the MSI subtype with their expression correlating positively with immune infiltrating cells.

### 3.9. Progression Free Survival

Based on our analysis of z-score using the Pan TCGA data from cBioPortal, we found that CRC samples with high *DYRK1A* expression levels have significantly lower progression free survival rate (Figure 13a); however, for other *DYRKs*, no significant change in PFS was observed (Figure 13b–d). These results are consistent with what described above and further support that the notion that *DYRK1A* expression in late-stage CRC tumors may reflect poor patient prognosis.

### 3.10. Overall Survival

To further confirm our finding from cBioPortal, we performed Kaplan–Meier analysis for overall survival (obtained from CRC samples using GEPIA2 web tool). Interestingly, CRC patients with higher *DYRK1A* expression showed significantly lower overall survival (Figure 14a), whereas other *DYRK* members did not show any significant correlation with overall survival (Figure 14b–f). This indicates that expression of *DYRK1A*, but not other *DYRK* members, can predict cancer survival in CRC.

## 4. Discussion

DYRKs were shown to be implicated in various human diseases, including cancer [10]. However, no study investigated the role of DYRKs specifically in colorectal cancer. In this study, using various web tools and large RNA sequence datasets derived from cancer patients, we analyzed specific expression levels of all DYRK family members in relation to CRC clinical features and patient survival outcome.

According to our analysis, *DYRK1A* is the predominantly expressed one in CRC; however, its physiological role in colonic epithelium is not yet fully elucidated. DYRK1A is well-known to be involved in various essential cellular pathways such as cell cycle regulation, specifically as a regulator for RNA polymerase II at specific genes loci that are involved in cell cycle, translation, and RNA processing [15,52]. This might give an insight into the predominance of DYRK1A in colon tissues.

It is noteworthy that, one of the main cellular functions of DYRK1A in neuronal cells is regulating cell cycle. Studies have shown that, in Down syndrome, when *DYRK1A* is overexpressed, it modulates the differentiation of neuronal cells, resulting in immature neurons via inhibiting the cell cycle [15]. Moreover, it was also found to inhibit cell cycle in cardiomyocytes [16], and it could be used as a therapeutic target to initiate pancreatic beta cells proliferation [53]. Together with DYRK1B, DYRK1A was found to have an inhibitory effect on cell cycle in COAD. One of the molecular mechanisms of class 1 DYRKs in cell cycle is that both were reported to phosphorylate cyclin D and induce its degradation and activate P27 [16,54,55].

One of the interesting findings of our analysis is that DYRK1A is the only member of the DYRK family to be associated with worse prognosis in CRC. It is significantly upregulated in later tumor stages, associated with lymph node and distant metastasis. Intriguingly, high expression of *DYRK1A* was associated with mucinous colorectal adenocarcinoma. It was reported that mucinous colorectal adenocarcinoma has a worse prognosis than non-mucinous colorectal adenocarcinoma [56]. Recently, it was reported that DYRK1A inhibition decreases metastases in tumors other than CRC, e.g., triple-negative breast cancer and glioblastoma cells [57,58]. Moreover, DYRK1A in endothelial cells was found to be involved in angiogenesis [59], which is an essential process for metastasis [60]. However, further studies should be conducted to investigate the role of DYRK1A in CRC metastasis.

Additionally, tumors with high DYRK1A level have been shown to have more potential of recurrence, which is associated with cancer stemness [61]. Interestingly, DYRK1A was recently reported to play a role in cancer stemness in oral squamous cell carcinoma. Therefore, inhibition of DYRK1A has been reported to reduce cancer stem cell phenotype [62]. These findings correlate with the clinical outcome; patients with high *DYRK1A* expression were found to have significantly lower PFS and lower OS.

The CRC is classified into major pathological molecular subtypes, (MSI) and (MSS). Usually, patients with MSI have a better prognosis than patients with MSS [63]. Here, we found that *DYRK1A*, *DYRK3* and *4* expression is higher in MSI, whereas *DYRK2* is higher in MSS. Both DYRK1A and DYRK2 have been reported to be involved in DNA-damage response, with depleted *DYRK2* cells found to have lower repair efficiency which correlates with our finding that *DYRK2* is higher in MSS subtype, which has a proficient DNA repair machinery [64].

Furthermore, DYRK1A was not reported to be involved in MMR pathway, it was only reported to act as a binding partner with ubiquitin ligase RNF169 that promote homologous recombination pathways after DNA double strand break [65]. Both DYRK1A and DYRK3 were found to activate the nicotinamide adenosine dinucleotide (NAD)-dependent deacetylase that inactivates TP53 under stressful conditions [38], because MSI has a higher mutational load and cellular stress, DYRK1A and 3 might be required to overcome cellular stress by activating SIRT1 and enhance the survival of cancer cells [38]. In metastatic dMMR–MSI-H CRC, immune checkpoint inhibitors (ICIs), specifically monoclonal antibodies targeting programmed cell death 1 (PD1) and cytotoxic T lymphocyte antigen 4 (CTLA4), increase survival, but pMMR–MSI-L CRC is mainly insensitive to existing ICIs [66].

Besides the variable genomic instability, MSI and MSS subtypes have differential immunogenic microenvironment. The MSI tumors were found to have a higher number of immune infiltrating cells than MSS [67]. In correlation with *DYRKs* expressions in CRC subtypes, *DYRK1A* and *3* significantly correlated with tumor infiltrating immune cells; this suggests that those two kinases can act as a prognostic marker for MSI and immunotherapy response. In the era of personalized medicine, finding the appropriate biomarkers that denote MSI-H is expected to predict the potential success of immunotherapy and to advance the management of CRC.

DYRK1B was found to have an anti-apoptotic effect in colon adenocarcinoma (COAD) and in global cancer predicted pathways (Figure 10). Additionally, it was found to inhibit the DNA damage response. In support of this role inhibiting DYRK1B in ovarian cancer cells was found to initiate apoptosis and to sensitize cancer cells to cisplatin (DNA damaging agent) [68], supporting the assumption that DYRK1B could act in a similar way in COAD. However, further studies are needed to support this hypothesis. Additionally, DYRK1B was found to activate PI3K/AKT pathway in COAD, its role in activating this pathway was reported in pancreatic and ovarian cancer cells [69].

DYRK2 is the only kinase that was found to be significantly downregulated in CRC, and was reported to have a tumor suppressor role in CRC [35]. DYRK2 could also be implicated in inhibiting both cell cycle and DNA damage response. It inhibits the cell cycle by inducing c-myc and c-Jun degradation [70]. However, DYRK2 role in inhibiting DNA damage response was not previously reported. In contrast it was reported to act as partner with ubiquitin ligase RNF8 to regulate DNA repair [64]. In our analysis, DYRK2 was found to be upregulated in MSS subtype that have a stable DNA repair machinery. Downregulation of DYRK2 impairs DNA repair efficiency [64]. Moreover, DYRK2 was reported to interact with P53, MDM2 and ATM [71]. DYRK2 is activated by ATM under DNA damage, this activation protects DYRK2 from MDM2-induced degradation and recruits DYRK2 to phosphorylate P53 and induce apoptosis. Such findings suggest that DYRK2 is a favorable kinase that suppresses the progression of CRC [71,72].

COAD-DYRK3 associated pathways included EMT and estrogen hormone pathway. DYRK3 was recently reported to be involved in glioblastoma multiform invasion and migration. Knocking down DYRK3 increased E-cadherin (epithelial marker) expression and decreased N-cadherin and Vimentin (Mesenchymal markers) expression in glioblastoma cancer cell [73]. Although in most cancers, DYRK4 contributes to cell cycle activation, its expression was not predicted in association with cancer pathways in COAD. Further studies should be conducted to understand the cellular role of DYRK4.

In general, *DYRK* genes were not mutated in studied CRC patient datasets, but variation in mRNA level was observed. We suggest that the variation in *DYRKs* expression in normal vs. tumor samples and in different tumor stages as in case of *DYRK1A* might not be linked to mutations, instead it might be due to epigenetic alterations. In correlation with *DYRK3* and *DYRK4* overexpression in tumors vs. normal, promotor methylation was significantly decreased in different tumor stages compared to normal. In COAD, there was upregulation in DYRK1B promoter methylation. *DYRK1B* had lower expression in COAD, however, it did not reach statistical significance. In READ *DYRK1B* was significantly less methylated compared to normal in READ stage 4. This correlates with its expression in READ, which is found to be non-significantly higher. For *DYRK1A* and *DYRK2*, there was no significant change in promotor methylation.

The current study provides an integrated comprehensive analysis of expression level, mutations, and immunological aspects of DYRKs in CRC carcinogenesis and progression. The study integrated pathological and clinical data to investigate the role of this family of dual kinases. Due to the limited availability of the molecular classification in most datasets and the very limited number of control cases in some analyses, our study has intrinsic limitations similar to other bioinformatics approaches. Furthermore, functional data such as enzyme activity status and ligand–receptor interactions have not been revealed in the TCGA and GTEx databases until recently. Therefore, we tried to overcome this challenge by studying protein network functional analysis, gene–gene, and protein–protein interaction analysis. To apply our results, more specific and extensive experimental work is required. Further studies should be conducted to understand the role of DYRKs in CRC, especially DYRK1A, hoping for promising new therapeutics to be developed targeting this kinase.

## 5. Conclusions

In the current study, we used publicly available data and web-based tools to investigate the status of DYRKs in colorectal cancer. We analyzed the expression in normal vs. tumor samples, then we investigated the expression of various DYRKs in different tumor stages and in CRC molecular and histological subtypes. Based on mRNA level, *DYRK1A* is the only member of DYRKs that was found to be upregulated in late tumor stages, with lymph node and distant metastatic stages. Additionally, it is the only kinase that was found to be associated with a worse progression free survival and overall survival when upregulated in patients‘ tumors. All DYRKs were found to be implicated in cancer-associated pathways which further indicate their involvement in cancer. No significant *DYRK* mutations were identified, and we suggested that *DYRK* expression variation in normal vs. tumor samples in different tumor stages might be because of epigenetics regulation. The majority of DYRKs showed a positive correlation with immune-infiltrating cells, which might suggest them as biomarkers for immunotherapy, namely, DYRK1A and DYRK3, which were upregulated in MSI subtypes. Our results reveal the role of DYRKs in CRC and provide deep insight into CRC biomarkers and potential novel therapies.

## Figures and Tables

**Figure 1 cancers-14-02034-f001:**
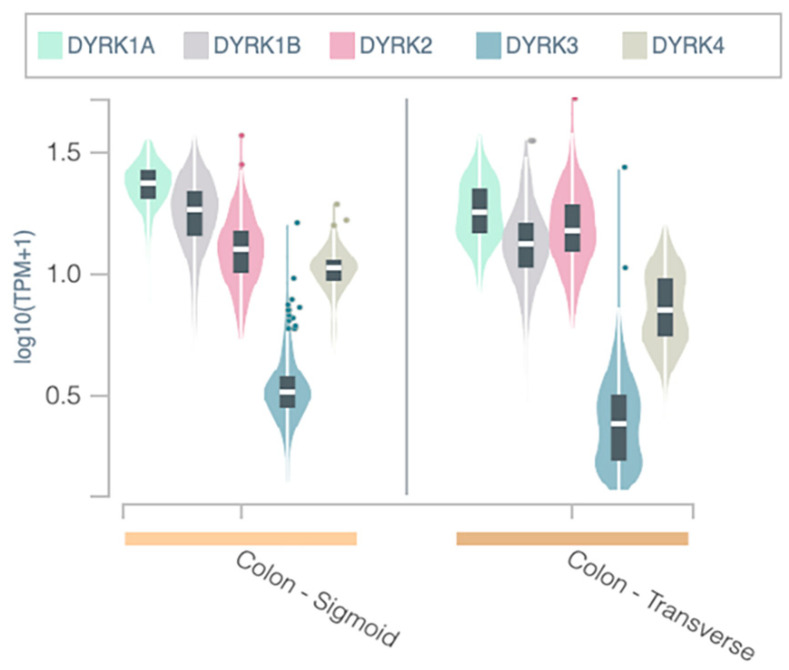
DYRKs expression in normal colon tissue: Using the GTEx portal web tool, DYRKs expression level from colon tissue was calculated based on RNAseq. TPM = transcript per million.

**Figure 2 cancers-14-02034-f002:**
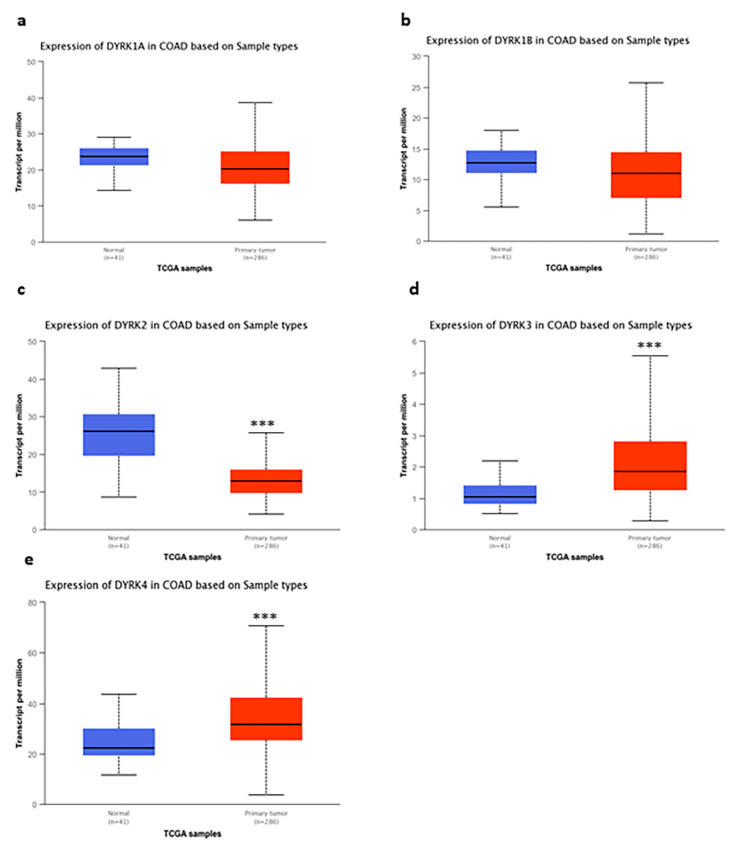
DYRKs expression in colon adenocarcinoma vs. normal tissues: using RNA seq dataset from TCGA data in UALCAN web tool we compared the expression of DYRKs in normal samples vs. tumor samples in colon adenocarcinoma. Student’s t-test was performed (**a**) *DYRK1A*, (**b**) *DYRK1B*, (**c**) *DYRK2*, (**d**) *DYRK3* and (**e**) *DYRK4*. *** *p* < 0.001.

**Figure 3 cancers-14-02034-f003:**
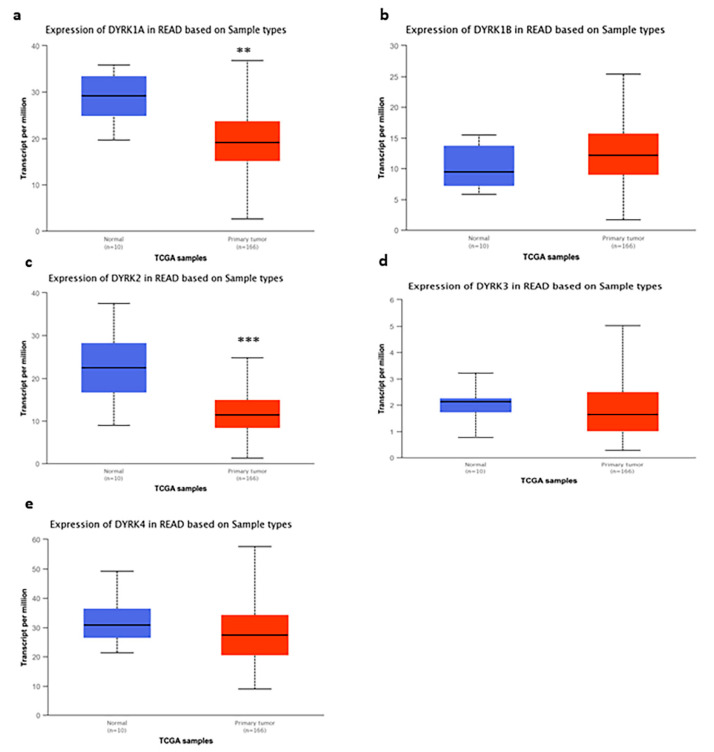
*DYRKs* expression in rectal adenocarcinoma vs. normal tissues: using RNA seq dataset from TCGA data in UALCAN web tool, we compared the expression of DYRKs in normal samples vs. tumor samples in colon adenocarcinoma. Student’s t-test was performed (**a**) *DYRK1A*, (**b**) *DYRK1B*, (**c**) *DYRK2*, (**d**) *DYRK3* and (**e**) *DYRK4*. ** *p* < 0.01 and *** *p* < 0.001.

**Figure 4 cancers-14-02034-f004:**
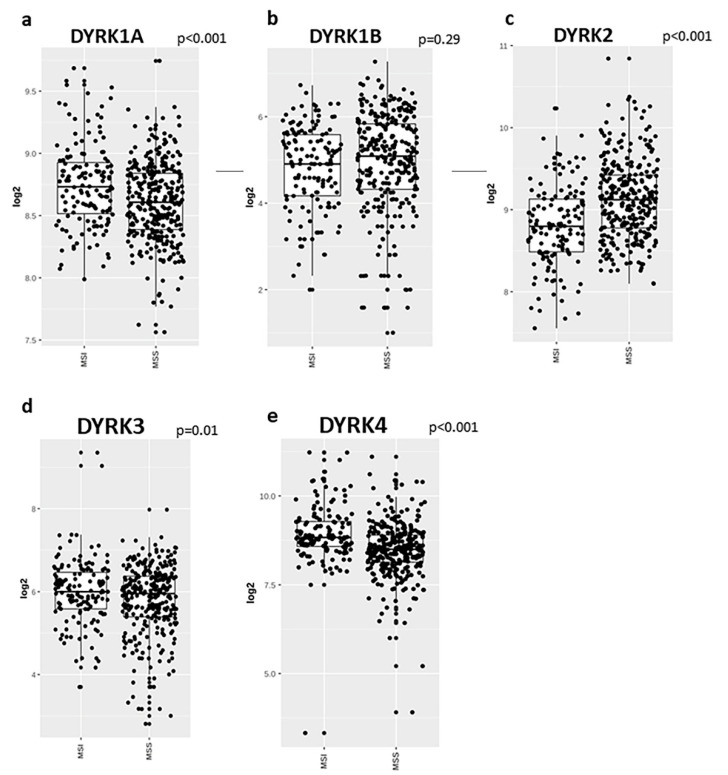
*DYRKs* expression level in colon cancer molecular subtypes. Using GENT2 web-based tool to represent data from NCBI GEO database, we query each DYRKs alone and chose colon cancer as tissue type and subtype as the targeted analysis. (**a**) *DYRK1A*, (**b**) *DYRK1B*, (**c**) *DYRK2*, (**d**) *DYRK3* and (**e**) *DYRK4*. Two sample T tests were conducted by the GENT2 web tool, *p* < 0.005 was considered as significant.

**Figure 5 cancers-14-02034-f005:**
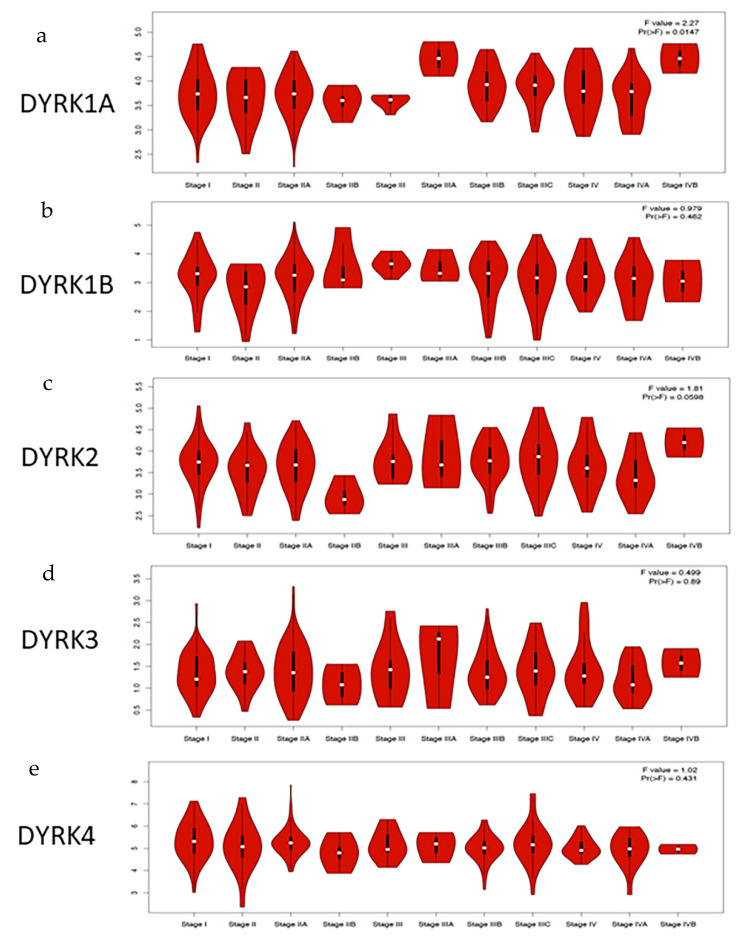
DYRKs expression level in CRC stages: using GEPIA2 web server that provide RNA seq from TCGA data set, DYRKs level in CRC stages were investigated. (**a**) *DYRK1A*, (**b**) *DYRK1B*, (**c**) *DYRK2*, (**d**) *DYRK3* and (**e**) *DYRK4*.

**Figure 6 cancers-14-02034-f006:**
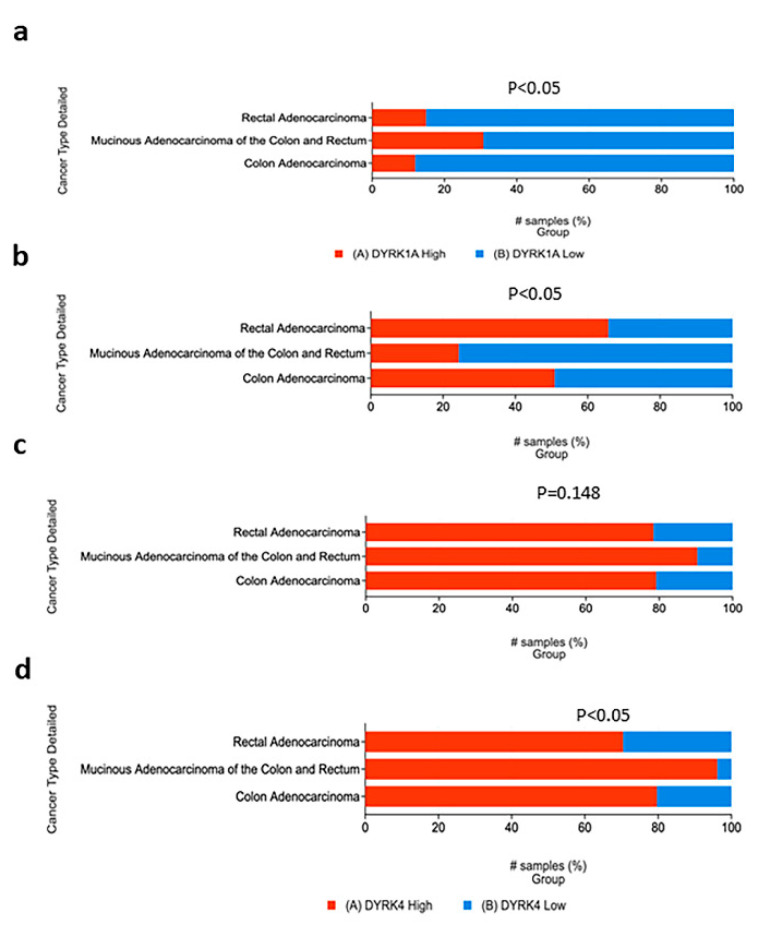
DYRKs status in CRC histological subtypes. PAN TCGA adenocarcinoma data set were used from cBioPortal. Samples were filtered based on mRNA z-score relative to normal samples (Log RNAseq V2 RSEM). Samples with mRNA z score = or >1 were considered high, and samples with mRNA z score <1 were considered low. (**a**) *DYRK1A*, (**b**) *DYRK1B*, (**c**) *DYRK3* and (**d**) *DYRK4*.

**Figure 7 cancers-14-02034-f007:**
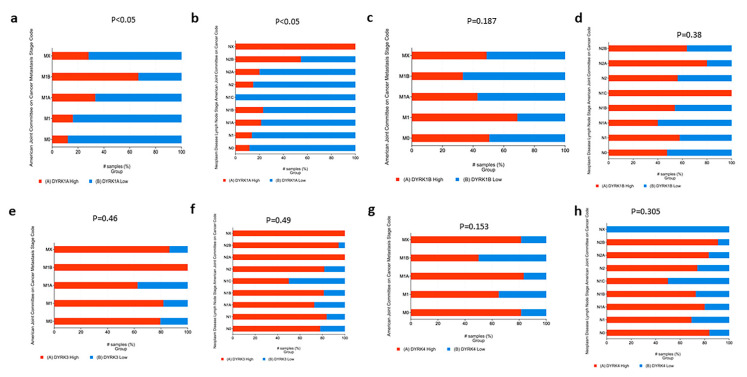
*DYRKs* status in metatsais and lymph node stages in CRC. PAN TCGA adenocarcinoma data set was used from cBioPortal. Samples were filtered based on mRNA z-score relative to normal samples (Log RNAseq V2 RSEM). Samples with mRNA z score = or >1 were considered high, and samples with mRNA z score <1 were considered low. (**a**) Lymph node stage for *DYRK1A*, (**b**) lymph node stage for *DYRK1A*, (**c**) metastasis stage for *DYRK1B*, (**d**) lymph node stage for *DYRK1B*, (**e**) metastasis stage for *DYRK3*, (**f**) lymph node stage for DYRK3, (**g**) metastasis stage for *DYRK4* and (**h**) lymph node stage for *DYRK4*.

**Figure 8 cancers-14-02034-f008:**
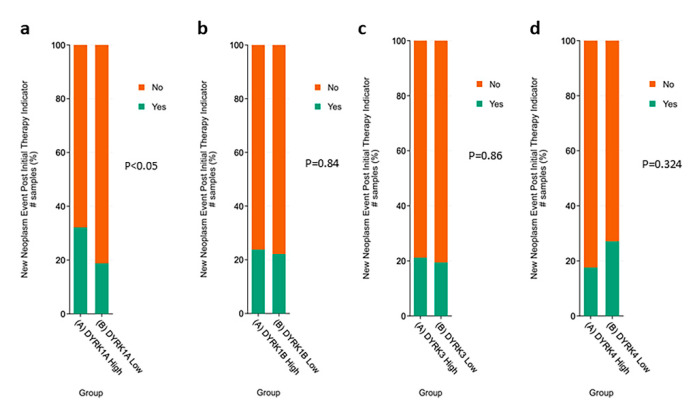
New newplasm event indicator. PAN TCGA adenocarcinoma data set were used from cBioPortal. Samples were filtered based on mRNA z-score relative to normal samples (Log RNAseq V2 RSEM). Samples with mRNA z score = or >1 were considered high, and samples with mRNA z score <1 were considered low. (**a**) *DYRK1A,* (**b**) *DYRK1B*, (**c**) *DYRK3* and (**d**) *DYRK4*.

**Figure 9 cancers-14-02034-f009:**
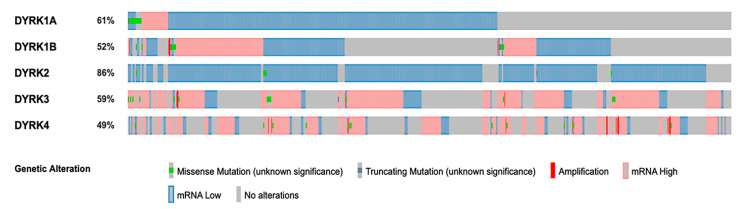
Mutations and copy number variation of *DYRKs* in CRC. Oncoprint represent DYRKs mutation and copy number in Pan TCGA adenocarcinoma data set obtained from cBioPortal. mRNA z-score relative to normal samples (Log RNAseq V2 RSEM) were chosen.

**Figure 10 cancers-14-02034-f010:**
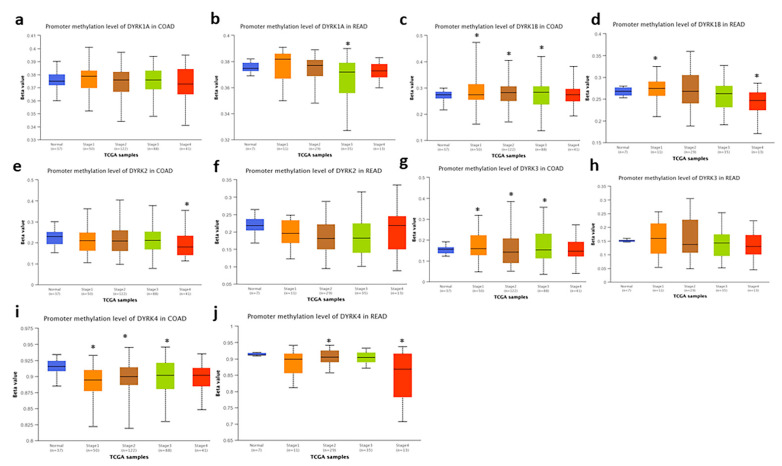
Promotor methylation level of *DYRKs* in COAD and READ. TCGA data set was obtained from UALCAN web tool. (**a**,**b**) *DYRK1A*, (**c**,**d**) *DYRK1B*, (**e**,**f**) *DYRK2*, (**g**,**h**) *DYRK3* and (**i**,**j**) *DYRK4*. * *p* < 0.05.

**Figure 11 cancers-14-02034-f011:**
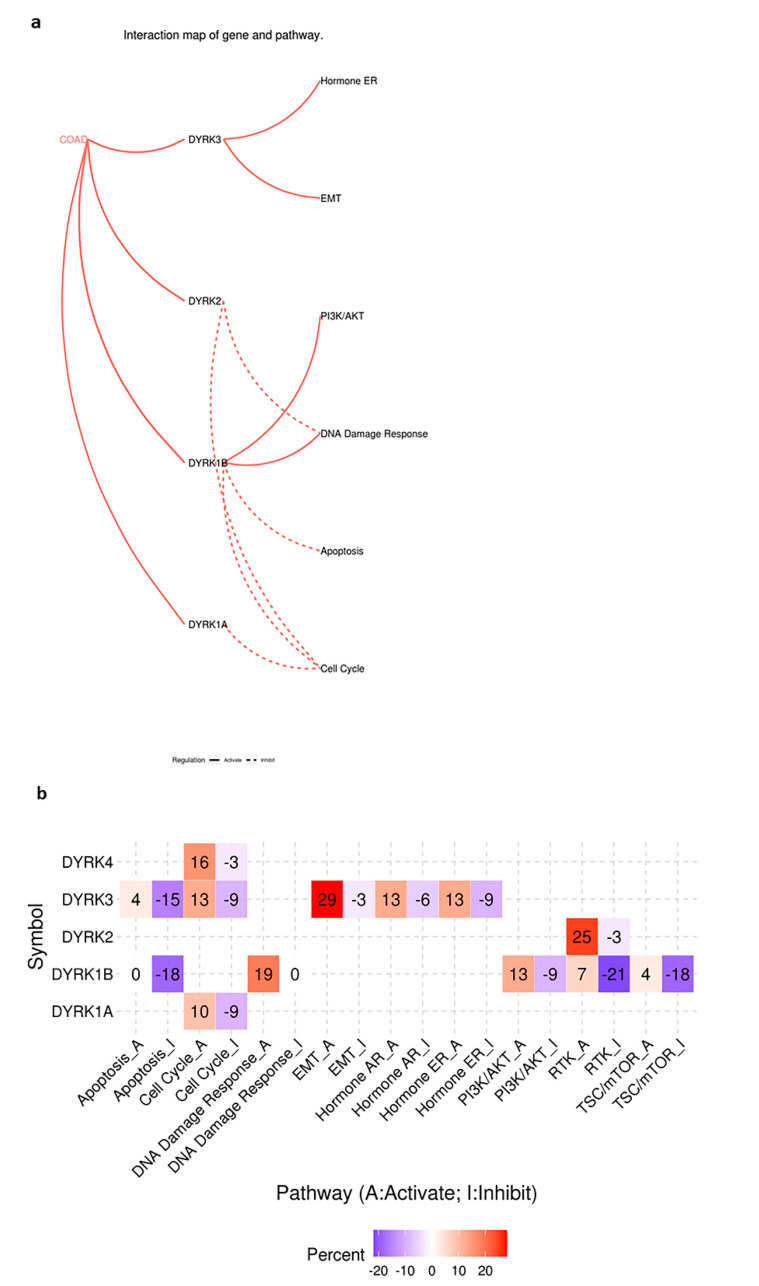
Cancer pathway activity of DYRKs predicted from GSCALite web tool. (**a**) Predicted pathways in COAD. (**b**) Predicted pathways in global cancers.

**Figure 12 cancers-14-02034-f012:**
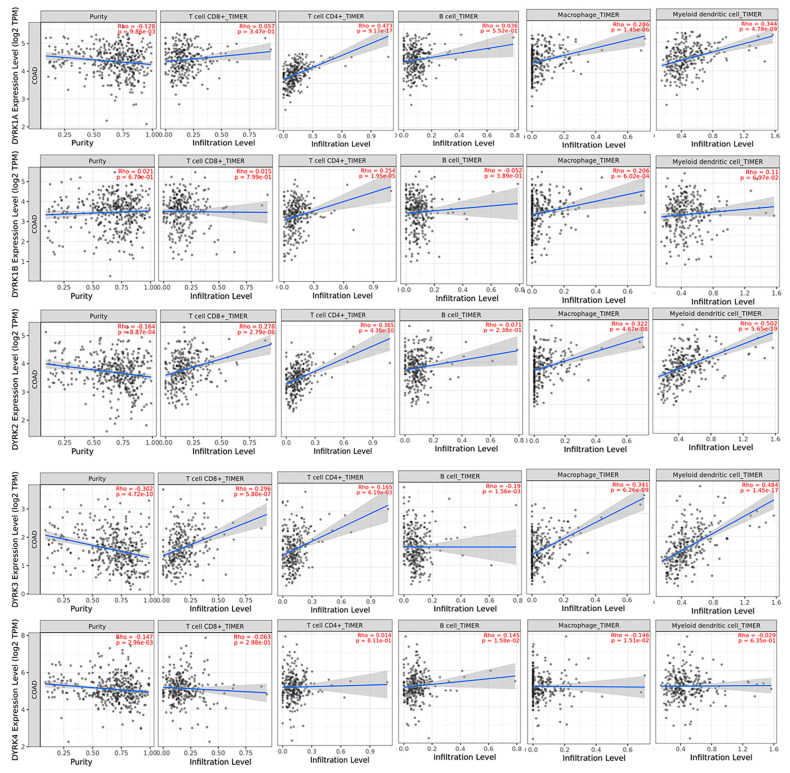
Spearman’s rank correlation coefficient of immune filtrating cells abundance and DYRKs expression in COAD obtained from TIMER 2 web tool.

**Figure 13 cancers-14-02034-f013:**
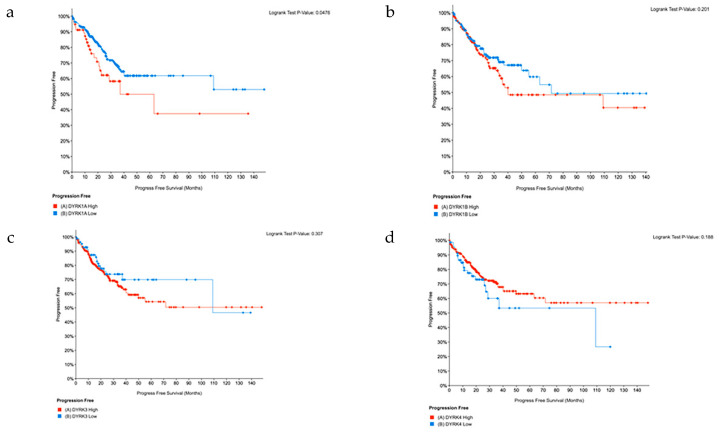
Progression-Free Survival. Pan TCGA adenocarcinoma data set were used from cBioPortal. Samples were filtered based on mRNA z-score relative to normal samples (Log RNAseq V2 RSEM). Samples with mRNA z score = or >1 were considered high and samples with mRNA z score <1 were considered low. (**a**) DYRK1A, (**b**) DYRK1, (**b**,**c**) DYRK3 and (**d**) DYRK4.

**Figure 14 cancers-14-02034-f014:**
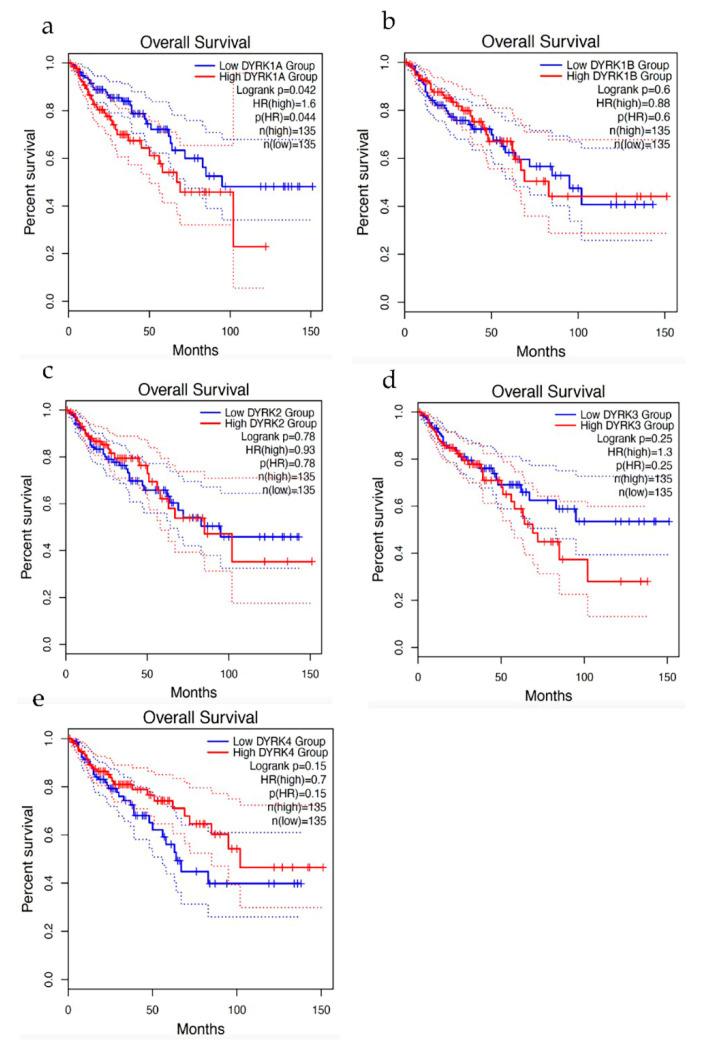
Overall Survival. Using GEPIA2 web server that provide RNA seq from TCGA data set, DYRKs overall survival for CRC patient were investigated. (**a**) DYRK1A, (**b**) DYRK1B, (**c**) DYRK2, (**d**) DYRK3 and (**e**) DYRK4.

## Data Availability

Not applicable.

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
