# Peer review of "A Bioinformatics Evaluation of the Role of Dual-Specificity Tyrosine-Regulated Kinases in Colorectal Cancer"

_cancers, 2022, doi:10.3390/cancers14082034_

Round 1

Reviewer 1 Report

The authors provide bioinformatics analysis of dual-specificity tyrosine kinases (DYRK) in colon cancer, and they take a bioinformatics approach to understand the roles of these classes of kinases in cancer. They show immune cell infiltration, which correlates with DYRK expression. The finding suggests that inhibition of DYRKs will be a helpful strategy for treating colon cancer. Some of the comments are below:

The authors could explain the method they employed in more detail and provide links if possible for replicating the analysis. In most cases, using multiple similar types of graphs may not be crucial for the article. The authors can consider moving some of the data set/figure panels into a supplementary file. The TIMER2 tool analysis is not explained in enough detail. Could the authors provide more details on how to interpret the data on the TIMER2 analysis?

For TCGA Data, how does promoter methylation correlate with expression? The mutations in the DYRKs are known to affect activity?

Should TIMIR2 be TIMER2? Line 116 and line159. The web address should be provided. The same should be done for the UALCAN. If the websites provide mechanisms to create permanent links, please provide those as well.

The coloring in the figure 2 legend might be off. Please make sure to use the same color/tone for example DYRK2

Were these genes (Figure 3) significantly up or downregulated in the overall analysis? Particularly in the colon adenocarcinoma? The statistics done in figure 3 should be mentioned in the legend

Figure 8 will be more apparent if the authors consider removing non-significant groups or using a table.

How were the plots generated for figure 10/11. Using cBioPortal/GEPIA2 or using other software?

Multiple PI3Ks are showing as P13K.

Author Response

The authors would like to thank the reviewer for the valuable comments that helped us a lot to improve the manuscript. Kindly find below our response to each comment and suggestion:

1. The authors could explain the method they employed in more detail and provide links if possible for replicating the analysis. In most cases, using multiple similar types of graphs may not be crucial for the article. The authors can consider moving some of the data set/figure panels into a supplementary file. The TIMER2 tool analysis is not explained in enough detail. Could the authors provide more details on how to interpret the data on the TIMER2 analysis?

Authors’ response:

-Links are now provided to each method plus a reference article to each method.

- In lines 193 -194, we explained that TIMER2 tool is basically a web tool that can calculate the correlation between an expression of a gene of interest with immune infiltrating cells by using multiple algorithms, we explained in the method part how we did the analysis.

-For data interpretation depending on the significant correlation (positive or negative) we predicted if DYRKs could contribute to the abundancy of immune infiltrating cells in tumor microenvironment, however, our predicted findings need further biological validation, as we stated in the limitation in the conclusion.

2. For TCGA Data, how does promoter methylation correlate with expression? The mutations in the DYRKs are known to affect activity?

Authors’ response:

Promotor methylation correlates with the expression of DYRK3 and 4 in COAD. TCGA data set were obtained from UALCAN web tool. For DYRK1A which was significantly downregulated in READ, promoter methylation correlate with the expression only in stage 3. For DYRK2 there was no significant change in promotor methylation although its significantly have low mRNA level in tumor samples, other epigenetics changes might contribute to this which need further studies and investigation in future. Mutations of DYRK is rarely found in CRC and has little effect on the level of expression, as mentioned in lines 560-562.  

3. Should TIMIR2 be TIMER2? Line 116 and line159. The web address should be provided. The same should be done for the UALCAN. If the websites provide mechanisms to create permanent links, please provide those as well.

Authors’ response:

Corrected. The links were provided in methodology part.

4. The coloring in the figure 2 legend might be off. Please make sure to use the same color/tone for example DYRK2

Authors’ response:

Well noted. We corrected the color tone. Now the color legend is typically matching the colors in the figure.

5. Were these genes (Figure 3) significantly up or downregulated in the overall analysis? Particularly in the colon adenocarcinoma? The statistics done in figure 3 should be mentioned in the legend

Authors’ response:

Not all DYRKs have the same expression pattern in colon adenocarcinoma. DYRK3 and 4 were upregulated significantly. DYRK2 was downregulated significantly. The statistical tests are now mentioned in figure 3 legend and methodology part for this tool. The t test was performed using a PERL script with Comprehensive Perl Archive Network (CPAN) module (the tool generates the p-value).

6. Figure 8 will be more apparent if the authors consider removing non-significant groups or using a table.

Authors’ response:

7. How were the plots generated for figure 10/11. Using cBioPortal/GEPIA2 or using other software?

Authors’ response:

For figure 10, we used the cBioportal. For figure 11, we used GEPIA2, both tools used TCGA datasets. The tools were mentioned in the corresponding figure legend.

8. Multiple PI3Ks are showing as P13K.

Authors’ response:  Well noted and corrected

Reviewer 2 Report

In this manuscript, the authors employed various web-based bioinformatics tools to illustrate the expression of DYRK isoforms in CRC. However, the results cannot fully support their conclusion, and the flow and the logic of the manuscript are unclear. The results are jumpy.

  1. From figures 3 and 4, the results were inconsistent. In conclusion, the authors mentioned DYRK1A and DYRK3 could be biomarkers for immunotherapy. The evidence they provided seems cannot fully support this statement. The authors should consider giving further information to support the conclusion. In addition, the author should determine which features of CRC would be associated with the suitability of immunotherapy and determine if these features would be associated with the expression of DYRK1A and DYRK3.
  2. The figure legends indicated P>0.05, P>0.01 and P>0.001. The authors should revise them.
  3. In figure 6, the label of DYRK1A is missing.
  4. In figure 8, the results were weak. Any other results to support the correlation between metastasis and DYRKs expression?
  5. The flow of the manuscript is not smooth. The authors should consider revising it extensively and reorganizing the manuscript.
  6. Apart from the survival curve, would it be possible to perform cox regression analysis to determine the hazard ratio of DYRKs?

Author Response

We would like to thank the reviewer for the valuable comments that helped us to substantially improve the manuscript. Kindly find below our response to each comment:

In this manuscript, the authors employed various web-based bioinformatics tools to illustrate the expression of DYRK isoforms in CRC. However, the results cannot fully support their conclusion, and the flow and the logic of the manuscript are unclear. The results are jumpy.

Authors’ response: We thank the reviewer for the valuable comment. We greatly modified the flow of the results. 

First, we demonstrated DYRK expression in CRC vs normal and in different molecular, histological subtypes and stages.

Then, we described the mutations and promotor methylation of DYRK. We described the effect of DYRK activity on cancer associated pathways. Then, the effect of DYRK expression on tumor- infiltrating immune cells.

Finally, we explored the effect of DYRK on PFS and OS.

For the conclusion we mentioned that these results have limitations as biological validation is needed, all results are obtained from patients’ dataset provided in the web tools, however, the value of our manuscript is to help elucidate or investigate the status of DYRKs in CRC because it was not investigated before, we are currently working on the biological validation of some aspects of DYRK.

  1. From figures 3 and 4, the results were inconsistent. In conclusion, the authors mentioned DYRK1A and DYRK3 could be biomarkers for immunotherapy. The evidence they provided seems cannot fully support this statement. The authors should consider giving further information to support the conclusion. In addition, the author should determine which features of CRC would be associated with the suitability of immunotherapy and determine if these features would be associated with the expression of DYRK1A and DYRK3. Authors’ response: 

    Figures 3 and 4 are showing the expression of DYRKs in CRC at two different anatomical sites; colon adenocarcinoma and rectal adenocarcinoma respectively, and as we did not show all DYRKs follow the same pattern of results in different anatomical parts.

    For the immunotherapy, we know that its early to suggest that DYRK1A and 3 could be a biomarker but because they were upregulated in MSI subtype (Immunotherapy sensitive subtype) and found to be significantly positively correlated with immune infiltering cells, we suggest that they could act as biomarkers. Certainly, further validation is required in the future. 
  2. The figure legends indicated P>0.05, P>0.01 and P>0.001. The authors should revise them.Authors’ response: Well noted and corrected.
  3. In figure 6, the label of DYRK1A is missing.Authors’ response: Well noted and corrected.
  4. In figure 8, the results were weak. Any other results to support the correlation between metastasis and DYRKs expression?Authors’ response: This figure was generated via cBioportal after filtering patients mRNA seq samples to low and high DYRKs expression. As shown samples with high DYRK1A are significantly associated with late metastasic stages, same for lymph node. The only results that support this is that DYRK1A was the only member of DYRKs found to be significantly upregulated in late tumor stages as shown in figure 6.
  5. The flow of the manuscript is not smooth. The authors should consider revising it extensively and reorganizing the manuscript.Authors’ response:  

    We have switched the order for some sections in the results part.  

    First, we demonstrated DYRK expression in CRC vs normal and in different molecular, histological subtypes and stages.

    Then, we described the mutations and promotor methylation of DYRK. We described the effect of DYRK activity on cancer associated pathways. Then, the effect of DYRK expression on tumor- infiltrating immune cells.

    Finally we explored the effect of DYRK on PFS and OS.

  6. Apart from the survival curve, would it be possible to perform cox regression analysis to determine the hazard ratio of DYRKs?Authors’ response: Hazard ratio (HR) and proportionate HR values are provided to each DYRK in figure 15 (overall survival) it is generated, using Cox regression, via GEPIA2 tool. In correlation with overall survival DYRK1A has the only significant HR with 1.6.  

Round 2

Reviewer 2 Report

The authors have already addressed most of the concerns, and they have made extensive efforts to improve the quality of the manuscript. The manuscript is acceptable for publication.

Some points that can improve the manuscript during the final editing process.

  1. Labels for stages in DYRK1A in Figure 6 are still missing.
  2. The clinical significance of DYRK1A and DYRK3 in CRD with MSI  was not addressed. 

Author Response

We would like to express our thanks to the reviewer for following up with valuable comments to improve the manuscript and to highlight its clinical significance. Kindly find below our response to the comments:

  1. Labels for stages in DYRK1A in Figure 6 are still missing.

Authors’ Response:

We apologize for this mistake. We added the missing axis title to figure 6.

  1. The clinical significance of DYRK1A and DYRK3 in CRD with MSI  was not addressed.

Authors’ Response:

In 754 and 755, we mentioned that the expression of DYRK1A, 3 and 4 is higher in MSI. We added a new statement to explain (lines 767-770) to introduce the clinical context. We added a new reference:

“In metastatic dMMR–MSI-H CRC, immune checkpoint inhibitors (ICIs), specifically monoclonal antibodies targeting programmed cell death 1 (PD1) and cytotoxic T lymphocyte antigen 4 (CTLA4), increase survival, but pMMR–MSI-L CRC is mainly insensitive to existing ICIs (66)”.

Then, we added a new sentence (line 776- 778) to the following paragraph to highlight the importance of finding a biomarker that may predict the MSI-H, hence the immunotherapy outcome:

“In the era of personalized medicine, finding the appropriate biomarkers that denote MSI-H is expected to predict the potential success of immunotherapy and to advance the management of CRC”.

Best Regards

(Corresponding Author)